# STRIDE: Structure and Embedding Distillation with Attention for Graph Neural Networks

## Abstract

Recent advancements in Graph Neural Networks (GNNs) have led to increased model sizes to enhance their capacity and accuracy. Such large models incur high memory usage, latency, and computational costs, thereby restricting their inference deployment. GNN compression techniques compress large GNNs into smaller ones with negligible accuracy loss. One of the most promising compression techniques is Knowledge Distillation (KD). However, most KD approaches for GNNs only consider the outputs of the last layers and do not consider the outputs of the intermediate layers of the GNNs. The intermediate layers may contain important inductive biases indicated by the graph structure and embeddings. Ignoring these layers may lead to a high accuracy drop, especially when the compression ratio is high. To address these shortcomings, we propose a novel KD approach for GNN compression that we call Structure and Embedding Distillation with Attention (STRIDE). STRIDE utilizes attention to identify important intermediate teacher-student layer pairs and focuses on using those pairs to align graph structure and node embeddings. We evaluate STRIDE on several datasets, such as OGBN-Mag and OGBN-Arxiv, using different model architectures, including GCNIIs, RGCNs, and GraphSAGE. On average, STRIDE achieves a $2.13\%$ increase in accuracy with a $32.3\times$ compression ratio on OGBN-Mag, a large graph dataset, compared to state-of-the-art approaches. On smaller datasets (e.g., Pubmed), STRIDE achieves up to a $141\times$ compression ratio with higher accuracy compared to state-of-the-art approaches. These results highlight the effectiveness of focusing on intermediate-layer knowledge to obtain compact, accurate, and practical GNN models. During the discussion phase, we will privately share the anonymized repo with reviewers and area chairs, and we will release it publicly upon acceptance.

## 1 Introduction

The rapid growth in the scale and complexity of real-world graphs, including social networks Wang et al. (2020), web graphs Web Data Commons (2024), knowledge graphs Wikipedia contributors (2024), e-commerce graphs GraphGeeks Lab (2024), and biological networks Koohi Esfahani et al. (2023) has driven significant advancements in Graph Neural Network (GNN) architectures, making them increasingly deeper and more expressive.

Recent studies examining neural scaling laws demonstrate notable accuracy improvements for GNNs by increasing depth, parameter count, and training dataset size Li et al. (2021); Yang et al. (2022b); Liu et al. (2025); Sypetkowski et al. (2023); Ma et al. (2022); Chen et al. (2024b); Airas & Zhang (2025). For instance, Liu et al. Liu et al. (2025) show clear performance gains with deeper and wider GNN models. Similarly, Sypetkowski et al. Sypetkowski et al. (2023) highlight enhanced performance in molecular graph tasks by employing larger models and richer pretraining datasets.

However, these performance gains come at significant costs, including increased computational complexity, memory usage, storage requirements, over-smoothing, and over-squashing, which complicate their practical deployment Liu et al. (2024); Sypetkowski et al. (2023); Di Giovanni et al. (2023). The growing demand for real-time inference further exacerbates these deployment challenges. Real-time applications such as autonomous vehicle point cloud segmentation Shi & Rajkumar (2020); Sarkar et al. (2023), high-energy physics data acquisition Shlomi et al. (2020), real-time recommendation

systems Liu et al. (2022), rapid image retrieval Formal et al. (2020), and spam detection Li et al. (2019) require extremely low inference latencies. Unfortunately, as GNN model complexity increases, inference latency escalates sharply, leading to substantial practical deployment barriers Zhou et al. (2021); Que et al. (2024); Tan et al. (2023); Huang et al. (2021); Kiningham et al. (2022). Consequently, compressing large GNNs into smaller, low-latency models without losing accuracy is now a key research goal.

Knowledge Distillation (KD) is a widely adopted model compression technique in which a compact *student* model is trained using supervision signals from a larger, well-performing *teacher* model Hinton et al. (2014). Although conventional KD can be applied directly to GNNs, it largely ignores structural properties inherent in graphs (e.g., Fitnets Romero et al. (2015) and Attention Transfer (AT) Zagoruyko & Komodakis (2017)). Hence, simply matching node embeddings or attention maps overlooks critical structural information in GNNs, limiting the effectiveness of these methods when directly applied to graph data.

Recently, Tian et al. Tian et al. (2025) identify three primary types of transferable knowledge in GNN distillation: *logits*, *structure*, and *embeddings*. Among these, logits-based distillation using soft-label predictions is straightforward, prompting recent research to focus on more advanced methods for transferring structural and embedding knowledge from teacher to student GNNs. Structural knowledge captures how nodes are interconnected and how the teacher network encodes graph topology Yang et al. (2020). Embedding knowledge mainly reflects node-level semantic relationships in the learned feature space He et al. (2022); Joshi et al. (2022). Early KD methods for GNNs primarily focused on preserving local graph structure. For example, LSP Yang et al. (2020) emphasizes the local structural alignment between the teacher and the student. Joshi et al. build on LSP by introducing GSP, which distills knowledge using all pairwise node similarities, and G-CRD, which preserves global topology via contrastive alignment of student and teacher node features Joshi et al. (2022). Later, GraphAKD He et al. (2022) directly distills embedding knowledge by forcing the student's node and class-level embeddings to match those of the teacher through adversarial training.

On the other hand, several studies have developed attention mechanisms for KD in GNNs, typically focusing on transferring knowledge from multiple teachers to a single student Wang et al. (2021); Zhang et al. (2022) or leveraging only embedding or structural features to enhance distillation.

Despite this progress, current KD methods for GNNs remain fundamentally limited by focusing mainly on final-layer embeddings, neglecting valuable information captured in intermediate layers Baxter (2000); Uselis & Oh (2025). Intermediate GNN layers encode distinct graph connectivity patterns and hierarchical structural relationships, which are critical for generalization. Ignoring these intermediate representations restricts the student's capability to learn deeper structural relationships, causing it to rely heavily on superficial mappings between node attributes and final-layer outputs, thus hindering generalization to unseen graph data. In particular, **it is essential to jointly leverage structural relationships, node embeddings, and intermediate-layer representations. These components collectively encode distinct yet complementary information about graph data.**

Unfortunately, aligning intermediate-layer representations poses a nontrivial challenge due to inherent architectural differences between teacher and student models. Typically, compressed student networks contain fewer layers, creating a mismatch in intermediate representations and preventing straightforward one-to-one alignment. Consequently, most existing GNN distillation methods avoid intermediate-layer alignment, limiting their ability to fully utilize the rich hierarchical information embedded within teacher layers Joshi et al. (2022); Kim et al. (2021); Jing et al. (2021); Wang et al. (2024); Huo et al. (2023); Wang & Yang (2024). Addressing this challenge represents an important research frontier in GNN distillation, motivating innovative approaches to dynamically align intermediate representations without relying on fixed-layer correspondences.

To address the shortcomings of existing KD methods for Graph Neural Networks, we propose *Structure and Embedding Distillation with Attention* (STRIDE). The core novelty of our approach lies in using a trainable attention mechanism to automatically identify and align the most informative pairs of intermediate layers, resolving the longstanding challenge of mismatched architectures in GNN distillation. Unlike prior methods that rely on explicit, fixed layer mappings, STRIDE enables flexible and dynamic distillation of structural and embedding information even when the teacher and student networks differ significantly in depth and architecture.

Specifically, STRIDE projects intermediate hidden representations from both teacher and student GNNs into a shared latent space, facilitating meaningful comparison across layers. Subsequently, a learned attention mechanism dynamically weighs the importance of aligning each potential pair of teacher and student layers based on their representational similarity and informativeness. By aligning embeddings and structural information at multiple intermediate layers, STRIDE encourages the student network to internalize the hierarchical reasoning and richer graph structures captured by the teacher, rather than relying solely on superficial input-output mappings (see Figure 1).

The main contributions of our work include:

1. We introduce STRIDE, the first attention-based GNN knowledge distillation framework capable of **simultaneously aligning structural and embedding** representations across **all intermediate layers**. Crucially, STRIDE accommodates substantial architectural differences (e.g., depth, hidden dimensions) between teacher and student models, without requiring explicit layer correspondence.

2. We develop a novel **attention-driven alignment mechanism**, enabling dynamic identification of critical teacher-student layer pairs. This facilitates effective knowledge transfer and improves the student's representational capability across diverse GNN configurations.

3. We provide extensive empirical validation of STRIDE using multiple widely-adopted benchmark datasets, such as OGBN-Mag Wang et al. (2020) and OGBN-Arxiv Wang et al. (2020), and various GNN architectures including GCNII Chen et al. (2020), RGCN Schlichtkrull et al. (2018), and GraphSAGE Hamilton et al. (2017). Our experiments demonstrate consistent performance improvements in accuracy and generalization across different degrees of model compression. On the large-scale OGBN-Mag dataset, our method outperforms state-of-the-art approaches by 2.13% in accuracy and achieves a $32.3\times$ compression ratio. On smaller datasets (e.g., Pubmed), STRIDE attains compression ratios as high as $141\times$ with higher accuracy relative to state-of-the-art baselines.

## 2 PROPOSED APPROACH

### 2.1 INTUITION AND MATHEMATICAL FOUNDATIONS

In this section, we first discuss the intuition behind STRIDE and introduce some of the mathematical definitions needed to explain it thoroughly. The mathematical notations are explained in the Appendix of the paper.

#### 2.1.1 SOFTKD INTUITION

In SoftKD Hinton et al. (2014), we compute two different losses. The first, $H(s_p, y)$, is a cross-entropy loss between the output student probability distribution and the ground truth labels. The other, $H(s_p, t_p)$, is a cross-entropy loss between the output student probability distribution and the output teacher probability distribution. The total loss is defined as:

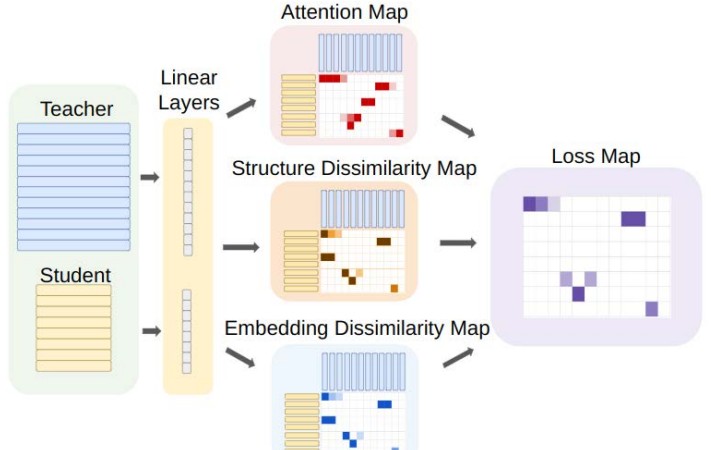

Figure 1: STRIDE generates an attention map using a trainable attention mechanism and a dissimilarity map using a trainable subspace projection. The loss matrix is an element-wise multiplication of the attention matrix and the dissimilarity matrix.

$$L_{KD} = H(s_p, y) + \alpha H(s_p, t_p) \tag{1}$$

Here, $\alpha$ is a hyper-parameter controlling how much the KD loss affects the total loss. The goal is to align the output student probability distribution with the output teacher probability distribution. The higher $H(s_p, t_p)$ is, the less aligned the student and teacher output probability distributions are.

### 2.1.2 STRIDE INTUITION

Similarly, STRIDE aims to incorporate alignment, but goes beyond final output alignment by focusing on intermediate layers, which encode valuable inductive biases. We pay special attention to the structural information within these representations. A key challenge arises from our goal to support arbitrary teacher-student architectures. Since their number of layers may differ, a direct layer-to-layer alignment is not feasible.

STRIDE solves this problem by identifying which teacher-student layer pairs are the most important to align via an attention mechanism. This mechanism works with an arbitrary number of teacher and student layers, which makes this approach amenable to any arbitrary teacher-student configuration. STRIDE also proposes a reprojection technique to account for the student and teacher networks having different hidden dimensions. The output of each hidden layer for both the teacher and student networks is projected into a standardized embedding dimension, which ensures that it will work with student and teacher networks of any embedding dimension (Figure 1).

As each layer represents unique semantic information, an important challenge is to ensure that each layer's feature map is not smoothed out by a single projection matrix. To this end, we use separate trainable linear layers for each hidden layer in both the teacher and student networks to ensure that we do not lose any valuable semantic information in the hidden layers. These trainable linear layers help us construct the three key components of STRIDE, which are the attention map, the structural dissimilarity map, and the embedding dissimilarity map. At a high level, the attention map tells us how important each teacher-student layer pair is, while the dissimilarity maps tell us how distant the feature maps of each teacher-student layer pair are in terms of both embedding and structure. The teacher-student layer pairs with higher attention scores are deemed as more important; STRIDE focuses on reducing their structural and embedding dissimilarity scores during training (Figure 1).

### 2.1.3 MATHEMATICAL FOUNDATION

Without loss of generality, we consider distilling a general Graph Convolutional Network (GCN) Kipf & Welling (2017), in which the output of the $l$-th layer is:

$$\mathbf{H}^{(l)} = \sigma(\hat{\mathbf{A}}\mathbf{H}^{(l-1)}\mathbf{W}^{(l)}) \tag{2}$$

Here, $\sigma$ is an activation function. The $\hat{\mathbf{A}} = \mathbf{D}^{-1/2}\mathbf{A}\mathbf{D}^{-1/2}$ is the normalized adjacency matrix, where $\mathbf{A}$ is the adjacency matrix and $\mathbf{D}$ is the diagonal degree matrix. The term $\mathbf{H}^{(l-1)} \in \mathbb{R}^{n \times d_{l-1}}$ represents the node feature matrix from the previous layer, and $\mathbf{W}^{(l)} \in \mathbb{R}^{d_{l-1} \times d_l}$ represents the trainable weight matrix of the current layer. Note that this formulation allows the hidden dimension to change among layers. In our method, we will denote the weight matrix for the $i$-th teacher layer as $\mathbf{W}_i^t$ and for the j-th student layer as $\mathbf{W}_j^s$.

For STRIDE, we are interested in the collection of intermediate feature representations from both the teacher and student networks. Let us define the teacher network $\mathcal{T}$ and the student network $\mathcal{S}$ as having $T_l$ and $S_l$ layers, respectively. We collect the *pre-activation* feature maps from each layer.

Let $\mathbf{T}_i \in \mathbb{R}^{n \times d_i^t}$ be the pre-activation output of the $i$-th layer of the teacher network, where $n$ is the number of nodes and $d_i^t$ is the feature dimension of that specific layer. Similarly, let $\mathbf{S}_j \in \mathbb{R}^{n \times d_j^s}$ be the pre-activation output of the $j$-th layer of the student network with output dimension $d_j^s$. Our goal is to distill knowledge from the set of teacher representations $\{\mathbf{T}_i\}_{i=1}^{T_l}$ to the set of student representations $\{\mathbf{S}_j\}_{j=1}^{S_l}$.

## 2.2 STRIDE MECHANISM

### 2.2.1 ATTENTION SCORES

The first step of STRIDE is to generate the attention matrix $\alpha \in \mathbb{R}^{T_l \times S_l}$. An element $\alpha_{ij}$ represents an "importance" score for the layer pair consisting of teacher layer $i$ and student layer $j$. We take the average of the feature maps along the node dimension to compute a mean node feature for every layer in both the teacher and student networks. We call these tensors $\mathbf{T}_a \in \mathbb{R}^{T_l \times d_t}$ and $\mathbf{S}_a \in \mathbb{R}^{S_l \times d_s}$. Then, we pass each layer in $\mathbf{T}_a$ through its own linear layer to create $\mathbf{T}_p \in \mathbb{R}^{T_l \times d_a}$, where $d_a$ is the embedding dimension of STRIDE. Similarly, we create $\mathbf{S}_p \in \mathbb{R}^{S_l \times d_a}$. We can finally generate $\alpha$ in the following manner:

$$\alpha = \text{softmax}\left(\frac{\mathbf{T}_p \mathbf{S}_p^T}{\sqrt{d_a}}\right) \tag{3}$$

The softmax is applied row-wise on the matrix product, such that for each teacher layer $i$, the sum of its attention scores across all student layers $j$ is equal to 1. This normalizes the importance scores from the perspective of a single teacher layer.

### 2.2.2 EMBEDDING DISSIMILARITY SCORES

The next step is to compute a pairwise embedding dissimilarity score for each teacher-student layer pair. Again, we project the features into $d_a$. For calculating the attention scores, we average over the node dimension before projecting, as our goal was to identify important layers. When calculating the pairwise embedding dissimilarity, we want to incorporate the per-node embeddings. So, we use a separate set of projection matrices. We use $\mathbf{P}_t \in \mathbb{R}^{d_t \times d_a}$ and $\mathbf{P}_s \in \mathbb{R}^{d_s \times d_a}$ to represent the projections. However, distance metrics are less semantically valuable if $d_a$ is high. To alleviate this problem, we define a trainable matrix $\mathbf{P} \in \mathbb{R}^{d_a \times d_a}$ to project all vectors into the subspace defined by the column space of $\mathbf{P}$. Since the rank of $\mathbf{P}$ can be less than $d_a$, distance metrics within the learned subspace can be more semantically valuable.

The final step is to average over the embedding dimension and then produce the embedding dissimilarity matrix $\mathbf{D}_{\text{emb}} \in \mathbb{R}^{T_l \times S_l}$. Its elements give the dissimilarity scores for each teacher-student layer pair. To calculate the embedding dissimilarity, we experiment with Euclidean and cosine distance, but Euclidean distance generally tends to perform better. The embedding dissimilarity score for a layer pair $(i, j)$ is a scalar obtained by aggregating the per-node differences. It can be represented as:

$$(\mathbf{D}_{\text{emb}})_{ij} = \frac{1}{n} \sum_{v=1}^{n} \left\| \left( \mathbf{T}_i[v,:] \mathbf{P}_t^{(i)} - \mathbf{S}_j[v,:] \mathbf{P}_s^{(j)} \right) \mathbf{P} \right\|_2^2 \tag{4}$$

Here, $\mathbf{T}_i[v,:] \in \mathbb{R}^{1 \times d_i^t}$ and $\mathbf{S}_j[v,:] \in \mathbb{R}^{1 \times d_j^s}$ are the feature vectors for node $v$ in teacher layer $i$ and student layer $j$, respectively. $\mathbf{P}_t^{(i)} \in \mathbb{R}^{d_i^t \times d_a}$ and $\mathbf{P}_s^{(j)} \in \mathbb{R}^{d_j^s \times d_a}$ are the layer-specific trainable projections. The aggregation is performed by averaging the squared Euclidean distance over all $n$ nodes.

### 2.2.3 STRUCTURAL DISSIMILARITY SCORES

Before calculating the loss, the final step is to compute a pairwise structural dissimilarity for each teacher-student layer pair. As usual, we first project teacher and student features to $d_a$ via a trainable linear projection. We use $\mathbf{E}_t \in \mathbb{R}^{d_t \times d_a}$ and $\mathbf{E}_s \in \mathbb{R}^{d_s \times d_a}$ to represent these trainable projections. After projecting to $d_a$ to find the structural dissimilarity, we simply use the G-CRD loss Joshi et al. (2022). We can also use other structure-aligning losses, such as LSP or GSP, but experimentally we find that using G-CRD produces the best results. We use the G-CRD loss to find the structural dissimilarity for every teacher-student layer pair to produce the structural dissimilarity matrix $\mathbf{D}_{\text{str}} \in \mathbb{R}^{T_l \times S_l}$. The structural dissimilarity score for a layer pair $(i, j)$ can be represented as:

$$(\mathbf{D}_{\text{str}})_{ij} = \phi(\mathbf{T}_i \mathbf{E}_t, \mathbf{S}_j \mathbf{E}_s) \tag{5}$$

where $\phi$ represents the structure-aligning loss of G-CRD.

Specifically, for each layer pair $(i, j)$, we adapt the G-CRD framework to create a contrastive loss. For a given anchor node $v$, its positive sample is its representation from the other network at the aligned layer (e.g., $T\_i$ for an anchor from $S\_j$), and negative samples are representations of other nodes from the same layer and network. The loss for a single layer pair is then the average of the contrastive loss over all nodes.

### 2.2.4 FINAL LOSS CALCULATION

To produce the final loss value, we first compute a total dissimilarity matrix $\mathbf{M} = \mathbf{D}_{\text{str}} + \mathbf{D}_{\text{emb}}$. We then multiply element-wise with the attention matrix $\alpha$ and take the mean to produce a single number that represents the STRIDE distillation loss, $L_{STRIDE}$.

$$L_{\text{STRIDE}} = \frac{1}{T_l S_l} \sum_{i=1}^{T_l} \sum_{j=1}^{S_l} \alpha_{ij} M_{ij} \tag{6}$$

The final loss is calculated as $L = H(s_p, y) + \beta L_{STRIDE}$ where $H(s_p, y)$ is the standard cross-entropy loss between the student's predictions and the ground truth labels. There is one important theorem to consider that proves $L_{STRIDE}$ distills valuable knowledge from the teacher network to the student network.

**Theorem 1 (STRIDE Cross-Layer Gradient Dependence)** *Let the STRIDE distillation loss be $L_{STRIDE}$, which is a function of the set of all teacher weight matrices $\{\mathbf{W}_i^t\}_{i=1}^{T_l}$ and student weight matrices $\{\mathbf{W}_j^s\}_{j=1}^{S_l}$. The gradient of the loss with respect to any student layer's weight matrix, $\mathbf{W}_j^s$, is functionally dependent on every teacher layer's weight matrix, $\mathbf{W}_i^t$. Formally:*

$$\frac{\partial L_{STRIDE}}{\partial \mathbf{W}_j^s} = f(\{\mathbf{W}_k^t\}_{k=1}^{T_l}, \{\mathbf{W}_l^s\}_{l=1}^{S_l}) \quad \forall i \in [1, T_l], \forall j \in [1, S_l] \tag{7}$$

*This holds even for teacher layers $i$ that are deeper than the student layer $j$ (i.e., $i > j$).*

**Intuitive Proof** The full proof involves a detailed expansion of the partial derivatives and is provided in the Appendix. The core intuition, however, is straightforward and relies on the chain rule through the attention mechanism.

1. The total STRIDE loss is a sum of losses for each teacher-student layer pair $(i, j)$, weighted by an attention score $\alpha_{ij}$. The loss for a single pair is $L_{ij} = \alpha_{ij} \cdot M_{ij}$, where $M_{ij}$ is the dissimilarity score.

2. Crucially, the **attention score $\alpha_{ij}$ is a function of the outputs** of teacher layer $i$ (denoted $\mathbf{T}_i$) and student layer $j$ (denoted $\mathbf{S}_j$).
$$\alpha_{ij} \propto \mathrm{g}(\mathbf{T}_i, \mathbf{S}_j) \tag{8}$$

3. The output of any teacher layer, $\mathbf{T}_i$, is a function of its weights, $\mathbf{T}_i = f_t(\mathbf{W}_1^t, \ldots, \mathbf{W}_i^t)$. Likewise, the student's output $\mathbf{S}_j$ is a function of its weights, $\mathbf{S}_j = f_s(\mathbf{W}_1^s, \ldots, \mathbf{W}_j^s)$.

4. Therefore, when calculating the weight update for $\mathbf{W}_j^s$ via the gradient $\frac{\partial L_{STRIDE}}{\partial \mathbf{W}_j^s}$, the chain rule must backpropagate through $\alpha_{ij}$. Since $\alpha_{ij}$ directly depends on the teacher's output $\mathbf{T}_i$, the gradient flowing to the student weight $\mathbf{W}_j^s$ will necessarily contain terms involving the teacher's weight $\mathbf{W}_i^t$.

This structure creates a computational graph where the teacher's weights from *every* layer influence the gradient of *every* student layer, thus proving the cross-layer dependency.

**Main Takeaway** Theorem 1 provides the theoretical justification for our core claim: STRIDE enables a richer, more comprehensive knowledge transfer than prior methods. The key insight is that our attention mechanism creates **direct gradient pathways from all teacher layers to all student layers**. This means a shallow student layer (e.g., layer 1) can receive immediate supervisory signals not just from the teacher's first layer, but also from its deepest layers (e.g., layer 5). This allows the student to learn how to represent complex, higher-order neighborhood information—a task usually reserved for deeper layers—much earlier in its own architecture. This ability to distill the teacher's entire representational hierarchy into a more compact student model is what leads to the significant gains in accuracy and generalization that we observe in our experiments.

## 3 EXPERIMENTS

### 3.1 EXPERIMENTAL SETUP

For our main experiments, we test STRIDE on two difficult datasets: OGBN-Mag and OGBN-Arxiv Hu et al. (2020); Wang et al. (2020). These datasets utilize temporal splitting to create validation and test sets that assess a model's ability to generalize to out-of-distribution data. For OGBN-Mag, we run experiments using RGCN Schlichtkrull et al. (2018) as the teacher and student models, and for OGBN-Arxiv, we run experiments using GAT Veličković et al. (2018) as the teacher model and GraphSAGE Hamilton et al. (2017) as the student model. This allows us to evaluate the effectiveness of STRIDE for different GNN architectures. It also allows us to assess if STRIDE can distill information between different types of GNN architectures. To further assess generalization, we also evaluate on smaller datasets, including Cora Mccallum et al. (2000), Citeseer Sen et al. (2008), Pubmed Namata et al. (2012), and NELL Carlson et al. (2010). In our experiments, we keep the

teacher model architecture and weights fixed and only modify the size of the student network. Each distillation method starts from the same set of weights and trains for the same number of epochs across 5 runs. For our baselines, we consider LSP Yang et al. (2020), GSP Joshi et al. (2022), G-CRD Joshi et al. (2022), Fitnets Romero et al. (2015), and Attention Transfer (AT) Zagoruyko & Komodakis (2017), which are most closely related to STRIDE. We run all experiments on a Tesla V100 GPU.

## 3.2 EXPERIMENTAL RESULTS

### 3.2.1 OUT-OF-DISTRIBUTION EVALUATION

As shown in Table 1 and Figure 2, STRIDE consistently outperforms state-of-the-art across various compression ratios on OGBN-Mag and OGBN-Arxiv. Notably, it achieves gains of 2.13% and 1.70% at $32.3\times$ and $16.1\times$ compression, respectively. Since these benchmarks target out-of-distribution generalization, the results demonstrate STRIDE's ability to produce student models with superior generalization compared to existing KD methods.

| Dataset | OGBN-Mag | OGBN-Arxiv |
|---|---|---|
| Teacher | RGCN (3L-512H-5.5M) | GAT (3L-750H-1.4M) |
| Student | RGCN (2L-32H-170K) | GraphSAGE (2L-256H-87K) |
| Teacher | 49.80 | 74.20 |
| Student | $44.23 \pm 0.47$ | $70.87 \pm 0.58$ |
| Fitnets | $44.87 \pm 0.84$ | $71.32 \pm 0.32$ |
| AT | $43.87 \pm 0.67$ | $71.04 \pm 0.48$ |
| LSP | $45.21 \pm 0.54$ | $71.47 \pm 0.45$ |
| GSP | $44.97 \pm 0.58$ | $71.97 \pm 0.64$ |
| G-CRD | $45.42 \pm 0.43$ | $71.87 \pm 0.56$ |
| STRIDE | $\mathbf{47.55 \pm 0.28}$ | $\mathbf{73.67 \pm 0.49}$ |
| Ratio | $32.3\times$ | $16.1\times$ |

Table 1: Average accuracies for a variety of large datasets. The results are based on the average of five trials, with each distillation method applied to the same set of student weights. The notation aL-bH-cM in the second and third rows means the model has "a" layers, a hidden dimension of "b", and "c" million trainable parameters.

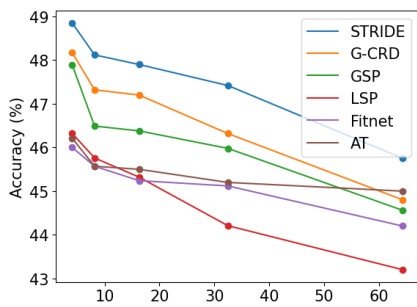

Figure 2: A comparison of KD methods applied to student models of different sizes trained on the OGBN-Mag dataset. The teacher model was the same as the one described in Table 1. The student model was a two-layer RGCN, and we varied the embedding dimension from 16 to 512 to induce this Pareto frontier.

### 3.2.2 ALIGNING INTERMEDIATE EMBEDDINGS IN STRIDE

To empirically prove that STRIDE aligns intermediate embeddings based on the attention matrix, we visualize the before and after training attention and dissimilarity maps in Figure 3. We train on OGBN-Mag and use a deeper teacher network of 5 layers and a hidden dimension of 512. The student network has 3 layers and a hidden dimension of 32. Our results show that dissimilarity scores are low where the attention scores are high and vice versa. This is in line with the intuition presented earlier in the STRIDE mechanism.

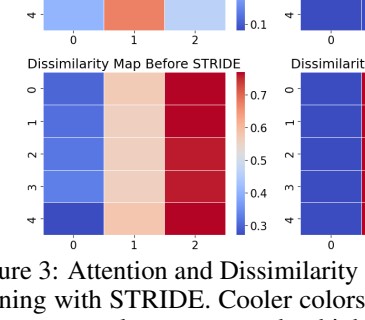

Figure 3: Attention and Dissimilarity maps before and after training with STRIDE. Cooler colors refer to lower scores and warmer colors correspond to higher scores.

**Deep GNNs**: We also test STRIDE on deep GNN architectures (e.g., GCNII Chen et al. (2020)). We test on Cora Mccallum et al. (2000), Citeseer Sen et al. (2008), Pubmed Namata et al. (2012), and NELL Carlson et al. (2010). Table 2 shows that STRIDE can distill these deep GCNIIs into shallower GCNIIs with higher accuracy compared to other distillation methods. At $27\times$ compression, STRIDE achieves a $3.5\%$ accuracy improvement. Even at a $141\times$ compression ratio, STRIDE matches the original teacher model's accuracy.

| Dataset | Cora | Citeseer | Pubmed | NELL |
|---|---|---|---|---|
| **Teacher** | GCNII (64L-64H) | GCNII (64L-64H) | GCNII (64L-64H) | GCNII (64L-64H) |
| **Student** | GCNII (4L-4H) | GCNII (4L-4H) | GCNII (4L-4H) | GCNII (4L-4H) |
| Teacher | 88.40 | 77.33 | 89.78 | 95.55 |
| Student | $73.87 \pm 0.42$ | $68.32 \pm 0.45$ | $87.87 \pm 0.45$ | $85.00 \pm 0.65$ |
| LSP | $75.07 \pm 0.55$ | $70.23 \pm 0.32$ | $88.07 \pm 0.45$ | $85.15 \pm 0.47$ |
| GSP | $78.22 \pm 0.31$ | $69.50 \pm 0.67$ | $89.19 \pm 0.55$ | $86.32 \pm 0.45$ |
| G-CRD | $83.45 \pm 0.45$ | $71.07 \pm 0.41$ | $89.66 \pm 0.48$ | $88.42 \pm 0.53$ |
| STRIDE | $\mathbf{84.27 \pm 0.32}$ | $\mathbf{72.00 \pm 0.30}$ | $\mathbf{89.89 \pm 0.31}$ | $\mathbf{92.02 \pm 0.64}$ |
| # St. Params | 5835 | 14910 | 2083 | 22686 |
| Ratio | $60.7\times$ | $33.5\times$ | $141.3\times$ | $27.7\times$ |

Table 2: Average accuracies for a variety of relatively smaller datasets. Each distillation method is applied to the same set of student weights.

### 3.2.3 IMPROVED WEIGHT INITIALIZATION FOR HIGHLY COMPRESSED NETWORKS

We find that for smaller datasets, information from the teacher network is mainly distilled into one layer of the student network, as shown in Figure 4. We hypothesize that smaller datasets lack complexity, allowing a single layer to capture most patterns.

To test this hypothesis, we first apply STRIDE to a student network of arbitrary size and then generate the attention map, $\alpha \in \mathbb{R}^{T_l \times S_l}$. The next step is to use a row-wise $argmax$ and find the student layer that has the most information distilled down to it. For example, in Figure 4, the selected layer for Cora would be the third student layer (index 2 in the Figure). We then instantiate a new one-layer network and copy over the weights from the identified layer (as indicated by the attention map $\alpha \in \mathbb{R}^{T_l \times S_l}$).

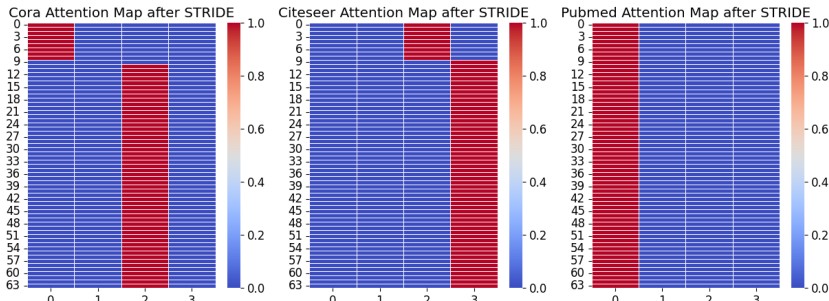

Figure 4: Attention maps for Cora, Citeseer, and Pubmed. Each color in the heatmap represents the importance score associated with that teacher-student layer pair. Warmer colors mean higher importance scores. It id apparent that most of the knowledge from the teacher layers is distilled into one student layer.

We then evaluate this new network on the test set and report the results in column 3 of Table 3. The first column of Table 3 represents the accuracies that we obtain after we train the new one-layer network for 1200 epochs; we compare this result to the accuracy obtained from training a one-layer network from random initialization, which we report in the second column of Table 3.

## 3.3 ABLATION STUDIES

We conduct a series of ablation studies in this subsection to further validate the effectiveness of STRIDE. Additional results and studies are provided in the Appendix.

### 3.3.1 BOOSTING PERFORMANCE BY ALIGNING BOTH STRUCTURE AND EMBEDDINGS

STRIDE is novel partly because it aligns both graph structure and node embeddings across teacher and student networks. To demonstrate the advantage of aligning both structure and embeddings, we compare STRIDE to variants that align only structure (S-STRIDE) or only embeddings (E-STRIDE). Results in Table 4 show that aligning both structures and embeddings is better than aligning just one of them.

### 3.3.2 IMPORTANCE OF ALIGNING INTERMEDIATE LAYERS

To prove that aligning intermediate layers is necessary for superior performance, we experiment with a variant of STRIDE, which we call *Modified* STRIDE, where we set $\alpha \in \mathbb{R}^{T_l \times S_l}$ to all zeros, but we set the bottom right value to 1. This indicates that we are only interested in the dissimilarity

| Dataset  | Initialized | Random Init | No Training |
|----------|-------------|-------------|-------------|
| Cora     | 80.35       | 73.59       | 65.36       |
| Citeseer | 70.21       | 68.15       | 54.20       |
| Pubmed   | 88.80       | 85.98       | 72.90       |

Table 3: Results for weight initialization experiment. These are all one-layer networks.

| Dataset  | OGBN-Mag            | OGBN-Arxiv         |
|----------|--------------------|--------------------|
| STRIDE   | $\mathbf{47.55 \pm 0.28}$ | $\mathbf{73.67 \pm 0.49}$ |
| E-STRIDE | $46.02 \pm 0.48$   | $71.32 \pm 0.53$   |
| S-STRIDE | $45.82 \pm 0.60$   | $71.48 \pm 0.57$   |

Table 4: STRIDE vs. S-STRIDE vs E-STRIDE. Teacher-student model configurations are in Table 1.

between the last layer node embeddings of the teacher and student models. The results in Table 5 prove that we gain accuracy by considering the outputs of intermediate layers for both teacher and student models. In this experiment, we start from the same set of initialized weights for both the STRIDE and modified STRIDE approaches.

### 3.3.3 IMPROVEMENTS DUE TO SUBSPACE PROJECTION

In Section 2, we introduced the concept of subspace projection as a way to alleviate issues caused by high-dimensional embedding spaces. While it is not needed for STRIDE to work, as Table 6 shows, it improves the results as the learned subspace projection matrix tends to be of lower rank than the embedding dimension. This indicates that we can project our feature maps into subspaces smaller than $R^{d_a}$, which increases the semantic value of the dissimilarity scores.

| Dataset         | OGBN-Mag           | OGBN-Arxiv         |
|-----------------|--------------------|--------------------|
| Student         | $44.46 \pm 0.54$   | $71.27 \pm 0.48$   |
| *Modified* STRIDE | $46.75 \pm 0.58$   | $71.76 \pm 0.52$   |
| STRIDE          | $\mathbf{47.58 \pm 0.31}$ | $\mathbf{73.59 \pm 0.45}$ |

Table 5: Comparing the modified STRIDE that only considers aligning the last layer node embeddings with STRIDE that considers intermediate layer node embeddings. Teacher/student model configurations are in Table 1.

| Dataset    | Subspace Projection | No Projection    |
|------------|---------------------|------------------|
| Cora       | $84.29 \pm 0.28$    | $83.78 \pm 0.35$ |
| Citeseer   | $72.05 \pm 0.35$    | $71.34 \pm 0.41$ |
| Pubmed     | $89.88 \pm 0.43$    | $88.76 \pm 0.53$ |
| NELL       | $91.95 \pm 0.58$    | $91.02 \pm 0.63$ |
| OGBN-Mag   | $47.54 \pm 0.32$    | $46.75 \pm 0.44$ |
| OGBN-Arxiv | $73.58 \pm 0.50$    | $73.00 \pm 0.53$ |

Table 6: Subspace projection impact. Teacher and student networks are the same as the ones in Tables 1 and 2.

### 3.3.4 NECESSITY OF A LINEAR LAYER FOR EACH HIDDEN LAYER

In our approach, we mentioned that each hidden teacher and student layer is assigned a linear layer for projection into $d_a$. This is because each layer represents its own $k$-hop neighborhood, and using just one linear layer would prove inadequate in capturing the full spectrum of essential semantic information contained within each layer. We run an experiment in which we use only one linear layer for the teacher and student projections. As Table 7 demonstrates, there is an accuracy drop compared to the situation where we use individual linear layers for the projection.

## 4 CONCLUSION

The ever-growing size and complexity of GNNs pose problems such as increased computational complexity, memory usage, storage requirements, over-smoothing, and over-squashing, which complicate their practical deployment for various applications such as real-time recommendation systems, spam detection, and rapid image retrieval. To address this difficulty, we propose an innovative solution known as Structure and Embedding Distillation with Attention (STRIDE). STRIDE employs an attention-based feature linking mechanism to identify important

| Dataset    | Multiple Linear Layers | One Linear Layer  |
|------------|------------------------|-------------------|
| Cora       | $84.29 \pm 0.28$       | $76.32 \pm 0.81$  |
| Citeseer   | $72.05 \pm 0.35$       | $69.00 \pm 1.07$  |
| Pubmed     | $89.88 \pm 0.43$       | $87.85 \pm 0.84$  |
| NELL       | $91.95 \pm 0.58$       | $85.67 \pm 0.72$  |
| OGBN-Mag   | $47.54 \pm 0.32$       | $45.02 \pm 0.67$  |
| OGBN-Arxiv | $73.58 \pm 0.50$       | $70.98 \pm 0.79$  |

Table 7: Linear layer per hidden layer effect. The teacher and student networks are the same as the ones described in Tables 1 and 2.

intermediate teacher-student layer pairs and focuses on aligning the node embeddings and graph structure of those pairs. This KD approach broadly outperforms existing KD approaches for GNNs over a wide variety of compression settings. It also works with both deep and shallow networks, and shows robust performance with different GNN architectures. On average, we achieve a $2.13\%$ increase in accuracy with a $32.3\times$ compression ratio on OGBN-Mag, a large graph dataset, compared to state-of-the-art approaches. On smaller datasets (e.g., Pubmed), STRIDE achieves a $141\times$ compression ratio with higher accuracy compared to the state-of-the-art methods.

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

# A APPENDIX

## A.1 LATENCY INCREASE BY NUMBER OF PARAMETERS

Figure 5 demonstrates how increasing the number of model parameters directly impacts inference latency. To visualize this trend, we plot the inference latency of a standard GCN model as we scale up its parameter count (e.g., by enlarging the embedding dimension) on the Flickr dataset. The figure reveals that as the model becomes more expressive and parameter-heavy, inference time rises substantially.

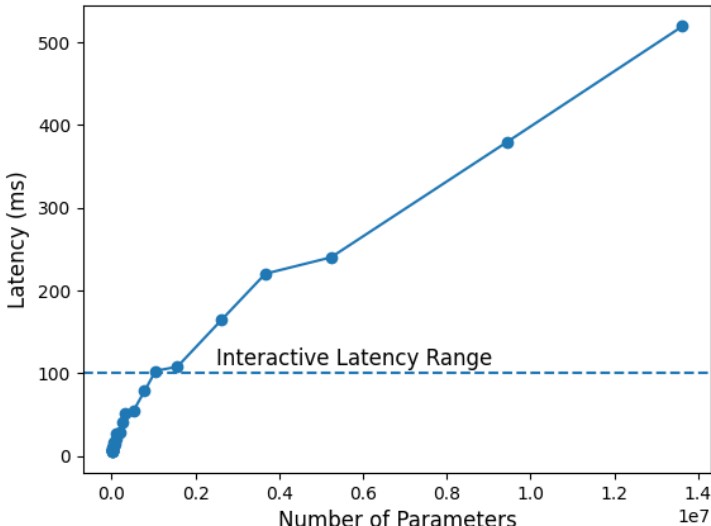

Figure 5: Inference latency of a standard GCN model architecture by increasing the number of parameters (e.g., embedding dimension) on the Flickr dataset. All tests were run on a Tesla V100 GPU with a batch size of 1.

## A.2 DATASET AND TEACHER NETWORK INFORMATION

Table 8 provides specifications of the datasets used in our experiments.

|            | # of Nodes | # of Edges | # of Features | # of Classes |
|------------|-----------|------------|---------------|--------------|
| Cora       | 2,708     | 10,556     | 1,433         | 7            |
| Citeseer   | 3,327     | 9,104      | 3,703         | 6            |
| Pubmed     | 19,717    | 88,648     | 500           | 3            |
| NELL       | 65,755    | 251,550    | 61,278        | 186          |
| Flickr     | 89,250    | 899,756    | 500           | 7            |
| OGBN-Mag   | 1,939,743 | 21,111,007 | 128           | 349          |
| OGBN-Arxiv | 169,343   | 1,166,243  | 128           | 40           |

Table 8: Specification of evaluated datasets.

## A.3 ABLATION STUDIES ON HYPERPARAMETERS

Two main hyperparameters need to be tuned when training STRIDE: the loss coefficient, $\beta$, and the STRIDE embedding dimension, $d_a$. We present the test accuracies across various values for $\beta$ and $d_a$ in Tables 9 and 10. These results show that a relatively lower $\beta$ and a higher $d_a$ tend to produce slightly better results. For all of our experiments, we used a $\beta$ of 10 and a $d_a$ of 256 for this reason.

## A.4 EUCLIDEAN VS. COSINE DISTANCE

In Section 2, we mention that we use the Euclidean distance metric instead of the cosine distance metric to generate the dissimilarity matrix, $M$. We present the results of this ablation in Table 11.

| Dataset | $\beta = 1$ | $\beta = 10$ | $\beta = 20$ | $\beta = 50$ |
|---|---|---|---|---|
| Cora | 86.92 | 84.71 | 86.19 | 85.64 |
| Citeseer | 73.20 | 74.33 | 71.82 | 69.82 |
| Pubmed | 89.03 | 89.97 | 88.32 | 88.56 |
| NELL | 90.86 | 88.73 | 90.14 | 91.32 |

Table 9: Ablation results for $\beta$.

| Dataset | $d_a = 64$ | $d_a = 128$ | $d_a = 256$ | $d_a = 512$ |
|---|---|---|---|---|
| Cora | 87.45 | 87.11 | 86.92 | 86.37 |
| Citeseer | 72.07 | 72.52 | 73.12 | 74.62 |
| Pubmed | 89.58 | 89.58 | 89.33 | 89.12 |
| NELL | 89.73 | 90.12 | 90.73 | 91.14 |

Table 10: Ablation results for $d_a$.

Cosine distance, which only considers the direction of vectors and not their magnitude, may disregard critical information contained in the magnitude of hidden representations. We hypothesize that preserving this information, as done by Euclidean distance, is important to its superior performance observed in our experiments.

| Dataset | Euclidean Distance | Cosine Distance |
|---|---|---|
| Cora | 87.33 | 82.81 |
| Citeseer | 73.43 | 70.98 |
| Pubmed | 89.58 | 87.32 |
| NELL | 91.24 | 85.66 |

Table 11: Test accuracies when using Euclidean vs. cosine distance for computing dissimilarity matrix $M$.

## A.5 DETAILS OF THE THEOREM AND THE PROOF

**Theorem 1 (STRIDE Cross-Layer Gradient Dependence)** *Let*

$$L_{STRIDE} = \mathbf{1}_{T_\ell}^\top (\boldsymbol{\alpha} \odot \mathbf{M}) \frac{\mathbf{1}_{S_\ell}}{S_\ell}, \qquad \mathbf{M} = \mathbf{D}_{emb} + \mathbf{D}_{str} \tag{6}$$

*be the distillation loss defined in Eq.(6) of the main paper, where $\boldsymbol{\alpha} \in \mathbb{R}^{T_\ell \times S_\ell}$ is the attention matrix of Eq.(3) and $\mathbf{M}$ collects the pair-wise embedding and structural dissimilarities of Eqs.(4)–(5). For every student layer $j \in [1, S_\ell]$ the gradient of $L_{STRIDE}$ with respect to the student weight matrix $W_j^s$ depends on every teacher weight matrix $W_i^t$ ($i = 1, \ldots, T_\ell$):*

$$\boxed{\frac{\partial L_{STRIDE}}{\partial W_j^s} = f\big(\{W_i^t\}_{i=1}^{T_\ell}, \{W_\ell^s\}_{\ell=1}^{S_\ell}\big)} \qquad \forall j.$$

*Consequently, gradients flow from* all *teacher layers—even those deeper than the student layer ($i > j$)—directly into $W_j^s$.*

**Proof:** For a single teacher–student layer pair $(i, j)$ define

$$L_{ij} := \alpha_{ij} M_{ij}, \qquad \alpha_{ij} = \frac{\exp z_{ij}}{\sum_{i',j'} \exp z_{i'j'}}, \tag{A1}$$

where the pre-soft-max score $z_{ij} = \frac{1}{n}\mathbf{1}_n^\top \underbrace{\hat{A} H_{i-1}^t W_i^t W_i^{pt}}_{T_i^p} (W_j^{ps})^\top (W_j^s)^\top (H_{j-1}^s)^\top \hat{A}^\top \mathbf{1}_n / n$ is a scalar obtained by taking the trace of the product of two length-$n$ vectors (so dimensions always match). $M_{ij}$ is the corresponding dissimilarity entry of $\mathbf{M}$.

By the product rule

$$\frac{\partial L_{ij}}{\partial W_j^s} = M_{ij}\frac{\partial \alpha_{ij}}{\partial W_j^s} + \alpha_{ij}\frac{\partial M_{ij}}{\partial W_j^s}. \tag{A2}$$

Because $\alpha_{ij}$ is a soft-max, $\frac{\partial \alpha_{ij}}{\partial z_{i'j}} = \alpha_{ij}(\delta_{ii'} - \alpha_{i'j})$. Applying the chain rule,

$$\frac{\partial \alpha_{ij}}{\partial W_j^s} = \sum_{i'=1}^{T_\ell}\frac{\partial \alpha_{ij}}{\partial z_{i'j}}\frac{\partial z_{i'j}}{\partial W_j^s} = \sum_{i'=1}^{T_\ell}\alpha_{ij}(\delta_{ii'} - \alpha_{i'j}) \tag{9}$$

$$\left[(H_{j-1}^s)^\top \hat{A}^\top \frac{\mathbf{1}_n}{n}\right]\left[\frac{\mathbf{1}_n^\top}{n} T_{i'}^p (W_j^{ps})^\top\right]. \tag{A3}$$

Each factor $T_{i'}^p = \hat{A} H_{i'-1}^t W_{i'}^t W_{i'}^{pt}$ contains the teacher weight matrix $W_{i'}^t$. Therefore $\partial \alpha_{ij}/\partial W_j^s$ depends on *every* $W_{i'}^t$. Both $D_{\text{emb},ij}$ and $D_{\text{str},ij}$ are functions of $T_i$ and $S_j$; their gradients w.r.t. $W_j^s$ pass through $T_i$ exactly once, so $\partial M_{ij}/\partial W_j^s$ also carries $W_i^t$.

The STRIDE loss is the average over all $(i,j)$: $L_{\text{STRIDE}} = \frac{1}{S_\ell}\sum_{i,j} L_{ij}$. Summing Eq.(A2) over $i$ preserves the dependence on every teacher weight appearing in (A3). Hence $\partial L_{\text{STRIDE}}/\partial W_j^s$ is a function of the whole set $\{W_i^t\}_{i=1}^{T_\ell}$. Since the argument holds for any student layer $j$, the gradient for *every* student layer jointly involves *all* teacher layers, completing the proof.

### A.6 DETAILED RELATED WORK

**KD for GNNs without Attention:** KD for GNNs is a relatively niche field that has been expanded recently. In LSP Yang et al. (2020), the authors attempt to align node embeddings between the student and teacher networks by maximizing the similarity between embeddings that share edges. Since only node embeddings between connected edges are aligned, this KD method preserves only local topology. Joshi et. al Joshi et al. (2022) extend LSP and propose two different KD algorithms: Global Structure Preserving Distillation (GSP) and Global Contrastive Representation Distillation (G-CRD). GSP extends LSP by considering all pairwise similarities among node features, not just pairwise similarities between nodes connected by edges. G-CRD implicitly preserves global topology by aligning the student and teacher node feature vectors via contrastive learning Oord et al. (2018). These works are examples of methods that focus on aligning structure as they use relationships between different nodes to transfer knowledge from the teacher to the student.

Mustad Kim et al. (2021) distills a large teacher GNN into a one-layer student GNN by minimizing a distance function between the student's final node embeddings and the teacher's final node embeddings. Some studies use adversarial training methods to distill knowledge from a teacher to a student network. GraphAKD He et al. (2022) treats the student network as a generator and trains a discriminator to distinguish between the final node embeddings of the student and teacher networks. Online Adversarial Distillation (OAD) Wang et al. (2024) trains multiple student models and trains a discriminator to distinguish between the outputs of different student models. More recent approaches, such as T2-GNN Huo et al. (2023), KDGCL Wang & Yang (2024), and SA-MLP Chen et al. (2024a), further advocate for utilizing embedding features of GNNs to improve KD in GNNs.

**Adapting CNN-based KD Approaches to GNNs:** There have been several KD approaches that have been applied to CNNs that the GNN community has tried to adapt to GNNs, including Fitnets Romero et al. (2015) and Attention Transfer (AT) Zagoruyko & Komodakis (2017). These methods both compute a distance metric, such as mean-squared error between the last layer node embeddings of the student and teacher networks, and do not take into account the adjacency matrix; therefore, these approaches can all be categorized as aligning only embeddings. Using attention to find similarities across student and teacher layers is a concept explored in CNNs Ji et al. (2021). However, the ideas from this work cannot be applied to GNNs because the feature-comparison operations are not applicable to graph data. GNNs need special consideration in this regard compared to CNNs due to the non-spatial and unstructured form of graph data.

**KD for GNNs via Attention:** Several works have constructed an attention mechanism for KD in GNNs; however, these approaches focus on distilling knowledge from multiple teachers to a single student. MSKD Zhang et al. (2022) uses an attention mechanism to assign weights to teacher networks in proportion to how much knowledge they should transfer to student networks. MulDE Wang et al. (2021) focuses on link prediction for knowledge graphs and uses a contrast attention mechanism to weigh soft labels from different teachers.

It is important to note that the above works only consider the node embeddings at the final layer of the teacher and student networks and aim to align them with one another in various ways. GeometricKD Yang et al. (2022a) aligns all teacher and student node embeddings, but it constrains the student and teacher networks to have the same number of layers to enforce a 1-1 correspondence between teacher and student layers. It then proceeds to align teacher layer $i$ with student layer $i$; this approach is inflexible as it severely constrains the student architecture.

Table 12 summarizes the main features of closely related work and how they are different from STRIDE. Unlike existing works, STRIDE aligns both structure and embeddings across all layers, without requiring strict architectural matching between teacher and student. This enables a richer transfer of hierarchical graph information and makes the approach applicable across diverse teacher-student architectures.

| Method | Aligns Structure | Aligns Embeddings | Number of Layers Considered |
|---|---|---|---|
| GraphAKD | × | ✓ | 1 |
| G-CRD | ✓ | × | 1 |
| LSP | ✓ | × | 1 |
| GSP | ✓ | × | 1 |
| Fitnets | × | ✓ | 1 |
| AT | × | ✓ | 1 |
| STRIDE | ✓ | ✓ | **All** |

Table 12: Comparison of various KD approaches with STRIDE

## A.7 SUMMARY OF NOTATIONS

In Section 2, we mathematically describe how STRIDE generates the attention matrix, $\alpha \in \mathbb{R}^{T_l \times S_l}$, the structural dissimilarity matrix, $Q$, and the embedding dissimilarity matrix, $D$, which is then used to calculate $L_{STRIDE}$. In Table 13, we provide a summary of all the mathematical notation used to describe STRIDE.

| Symbol | Meaning / Shape |
|---|---|
| *Graph primitives* | |
| $n$ | Number of nodes in the input graph |
| $\mathbf{A} \in \{0,1\}^{n \times n}$ | Binary adjacency matrix |
| $\mathbf{D} = \mathrm{diag}(d_1, \ldots, d_n)$ | Degree matrix ($d_i = \sum_j A_{ij}$) |
| $\hat{\mathbf{A}} = \mathbf{D}^{-1/2}\mathbf{A}\mathbf{D}^{-1/2}$ | Symmetric normalised adjacency (Eq. 2) |
| $\mathbf{1}_n \in \mathbb{R}^n$ | Vector of ones (all entries = 1) |
| *Index sets and dimensions* | |
| $T_\ell,\ S_\ell$ | Number of layers in teacher / student networks |
| $i \in \{1, \ldots, T_\ell\}$ | Teacher-layer index |
| $j \in \{1, \ldots, S_\ell\}$ | Student-layer index |
| $d_t,\ d_s$ | Hidden dimension of a teacher / student layer |
| $d_t^{(i)},\ d_s^{(j)}$ | Output dimension of teacher layer $i$ / student layer $j$ |
| $d_l$ | Hidden dimension at generic layer $l$ |
| $d_a$ | STRIDE latent dimension for projections / attention |
| *Layer outputs and projections* | |
| $\mathbf{H}^{(l)} \in \mathbb{R}^{n \times d_l}$ | Node-feature matrix at layer $l$ |
| $\mathbf{T}_i = \mathbf{H}_i^t \in \mathbb{R}^{n \times d_t}$ | Pre-activation output of teacher layer $i$ |
| $\mathbf{S}_j = \mathbf{H}_j^s \in \mathbb{R}^{n \times d_s}$ | Pre-activation output of student layer $j$ |
| $\mathbf{T}_i^p = \hat{\mathbf{A}}\mathbf{H}_{i-1}^t W_i^t W_i^{pt}$ | Projected teacher representation (Eq. 3) |
| $\mathbf{S}_j^p = \hat{\mathbf{A}}\mathbf{H}_{j-1}^s W_j^s W_j^{ps}$ | Projected student representation (Eq. 3) |
| $\mathbf{T}_i^p,\ \mathbf{S}_j^p \in \mathbb{R}^{d_a}$ | Mean-pooled projected representations |
| *Trainable weight matrices* | |
| $W_i^t \in \mathbb{R}^{d_t^{(i-1)} \times d_t^{(i)}}$ | GNN weight matrix of teacher layer $i$ |
| $W_j^s \in \mathbb{R}^{d_s^{(j-1)} \times d_s^{(j)}}$ | GNN weight matrix of student layer $j$ |
| $W_i^{pt} \in \mathbb{R}^{d_t \times d_a}$ | Teacher projection for attention (layer $i$) |
| $W_j^{ps} \in \mathbb{R}^{d_s \times d_a}$ | Student projection for attention (layer $j$) |
| $P_t \in \mathbb{R}^{d_t \times d_a},\ P_s \in \mathbb{R}^{d_s \times d_a}$ | Embedding dissimilarity projections |
| $E_t \in \mathbb{R}^{d_t \times d_a},\ E_s \in \mathbb{R}^{d_s \times d_a}$ | Structural dissimilarity projections |
| $\mathbf{P} \in \mathbb{R}^{d_a \times d_a}$ | Shared low-rank sub-space projection (§3) |
| *Attention and dissimilarity tensors* | |
| $\alpha_{ij}$ | Attention score for teacher layer $i \leftrightarrow$ student layer $j$ |
| $\boldsymbol{\alpha} \in \mathbb{R}^{T_\ell \times S_\ell}$ | Full attention matrix (Eq. 3) |
| $z_{ij}$ | Scalar pre-soft-max compatibility score for pair $(i,j)$ |
| $D_{ij}^{\mathrm{emb}}$ | Pairwise *embedding* dissimilarity (Eq. 4) |
| $D_{ij}^{\mathrm{str}}$ | Pairwise *structural* dissimilarity (Eq. 5) |
| $\mathbf{D}_{\mathrm{emb}},\ \mathbf{D}_{\mathrm{str}} \in \mathbb{R}^{T_\ell \times S_\ell}$ | Two dissimilarity matrices |
| $M_{ij} = D_{ij}^{\mathrm{emb}} + D_{ij}^{\mathrm{str}}$ | Total dissimilarity for pair $(i,j)$ |
| $\mathbf{M} = \mathbf{D}_{\mathrm{emb}} + \mathbf{D}_{\mathrm{str}}$ | Total dissimilarity matrix |
| *Losses, operators and hyper-parameters* | |
| $L_{ij} = \alpha_{ij} M_{ij}$ | STRIDE loss contribution of a single layer pair |
| $L_{\mathrm{STRIDE}}$ | Global STRIDE distillation loss (Eq. 6) |
| $\mathcal{H}(\cdot, \cdot)$ | Cross-entropy loss (logit supervision) |
| $\sigma(\cdot)$ | Element-wise activation function |
| $\phi(\cdot, \cdot)$ | G-CRD structural contrastive loss (Eq. 5) |
| $\odot$ | Hadamard (element-wise) product |
| $\beta$ | Trade-off coefficient in the total loss (Eq. 7) |

Table 13: Comprehensive notation used throughout the STRIDE paper. Bold uppercase symbols denote matrices, bold lowercase symbols denote vectors, and plain symbols denote scalars unless stated otherwise. Dimensions are provided where applicable.