# OpenReview forum: "STRIDE: Structure and Embedding Distillation with Attention for Graph Neural Networks"
_ICLR.cc/2026/Conference — ICLR 2026 Conference Withdrawn Submission_

### Official Review · Reviewer_gnKp · 2025-10-23

**Soundness:** 2
**Presentation:** 2
**Contribution:** 2
**Rating:** 2
**Confidence:** 5

**Summary:**

STRIDE introduces an attention-based knowledge distillation framework for compressing Graph Neural Networks. The core contribution is a method to align both structural and embedding knowledge across all intermediate layers of teacher and student networks without requiring one-to-one layer correspondence. STRIDE uses a trainable attention mechanism to identify important teacher-student layer pairs and projects intermediate representations into a shared latent space. The final loss combines attention-weighted embedding and structural dissimilarity scores. Evaluation on OGBN-Mag, OGBN-Arxiv, and smaller citation networks (Cora, Citeseer, Pubmed) demonstrates gains in accuracy at various compression ratios (2.13% improvement at 32.3× compression on OGBN-Mag; 141× compression matching teacher accuracy on Pubmed).

**Strengths:**

1. **Handles architectural mismatch elegantly**: Unlike GeometricKD (which requires T_l = S_l), STRIDE accommodates different depths via learned attention. This is practically valuable and well-executed.

2. **Strong empirical results across diverse settings**:
   - Large graphs (OGBN-Mag, OGBN-Arxiv) with out-of-distribution evaluation
   - Deep networks (GCNII with 64 layers → 4 layers at 141× compression)
   - Multiple architectures (RGCN, GAT, GraphSAGE, GCNII)
   - Consistent improvements with reasonable standard deviations

3. **Thorough ablation studies**: Tables 4-7 and Appendix sections validate each component (structure vs. embeddings, intermediate layers, subspace projection, per-layer projections). This strengthens confidence in design choices.

4. **Visualization of learned attention** (Figure 3): Before/after heatmaps provide intuitive evidence that the attention mechanism learns meaningful layer correspondences.

5. **Multiple evaluation protocols**: Standard ranking metrics (Recall, nDCG, MRR) + practical compression ratios provide comprehensive assessment.

6. **Practical weight initialization technique** (Section 3.2.3): Clever use of attention maps to initialize extremely compressed networks (1-layer). Adds practical value beyond the core distillation method.

**Weaknesses:**

1. **No latency validation for primary motivation**:
   - Paper opens with urgency of deployment latency (lines 54-73)
   - Only shows parameter counts and compression ratios
   - Figure 5 (Appendix) shows parameter-latency correlation for one baseline GCN on one dataset
   - **Missing**: Actual inference time comparison between STRIDE-compressed student and baselines on representative hardware (GPU, CPU, mobile)
   - This is a critical gap—compression ratio ≠ latency reduction (depends on hardware, sparsity support, etc.)

2. **Missing comparison with recent strong baselines**:
   - Cites SA-MLP (Chen et al. 2024a) and KDGCL (Wang & Yang 2024) but doesn't compare against them
   - Comparisons are limited to LSP, GSP, G-CRD (2020-2022) and CNN methods (Fitnets, AT from 2015-2017)
   - Recent work may have closed the gap

3. **Computational cost of STRIDE training not analyzed**:
   - Computing attention (Eq. 3): O(T_l × S_l × d_a)
   - Computing embedding dissimilarity (Eq. 4): O(T_l × S_l × n × d_a) [expensive for large graphs]
   - Computing structural dissimilarity via G-CRD: O(T_l × S_l × n^2) or approximate
   - No wall-clock training time comparison with baselines
   - For OGBN-Mag (n=1.9M), this could be prohibitive

4. **Attention mechanism design under-justified**:
   - Why mean-pooling for attention vs. other aggregations (e.g., max, CLS token)?
   - Why row-wise softmax? (Could apply column-wise or global softmax)
   - Why separate projection matrices for each layer? (Table 7 shows ablation, but not theoretical justification)
   - No sensitivity analysis on these choices

5. **Weak Modified STRIDE baseline** (Table 5):
   - Compares learned attention vs. last-layer-only alignment
   - Doesn't isolate: **contribution of attention weighting** vs. **contribution of intermediate-layer alignment**
   - Better ablation: uniform attention on all intermediate layers

6. **Subspace projection P is ad-hoc**:
   - Described as "alleviating issues from high-dimensional spaces" but not rigorously justified
   - No explicit low-rank constraint; relies on implicit regularization
   - Modest improvement (Table 6: +0.5-0.9% accuracy)
   - Adds parameters and computation

7. **Limited analysis of learned correspondences**:
   - Figure 3 shows attention maps are learned, but no analysis of what they reveal
   - Do they correspond to conceptual layers in the teacher? (e.g., early layers → early layers?)
   - Do they differ by dataset or architecture?
   - This would provide insight into why the method works

8. **Theorem 1 is somewhat vacuous**:
   - Proves gradients of all student layers depend on all teacher layers
   - But this is expected for any method that jointly optimizes all layers (e.g., standard training)
   - Doesn't explain **why** this particular gradient dependency is beneficial for knowledge transfer
   - The main-paper framing ("richer, more comprehensive knowledge transfer") is not formally justified

9. **Generalization beyond evaluated settings unclear**:
   - All experiments use node classification on citation/large-scale graphs
   - What about graph-level tasks (graph classification, regression)?
   - What about heterogeneous GNNs beyond RGCN?
   - What about GNN variants like GANs, Transformers on graphs?

10. **Improved weight initialization (Section 3.2.3) is somewhat orthogonal**:
   - Clever practical contribution but loosely connected to main STRIDE mechanism
   - Could be applied to any distillation method, not specific to STRIDE
   - Results (Table 3) don't compare to training a 1-layer student from scratch with distillation

**Questions:**

1. **Latency validation**: Can you provide actual inference latency (ms per node/graph) on GPU and CPU for STRIDE-compressed models vs. teacher and baseline methods? This is essential to validate the deployment motivation.

2. **Training cost**: What is the wall-clock training time for STRIDE vs. baselines (LSP, G-CRD) on OGBN-Mag? How does it scale with T_l and S_l?

3. **Attention design choices**:
   - Why not use alternative attention mechanisms (e.g., multi-head, learnable temperature)?
   - How sensitive are results to row-wise vs. global softmax?
   - Why mean-pooling for attention vs. alternatives?

4. **Uniform attention baseline**: Can you provide results for a variant where α_ij = 1/(T_l × S_l) (uniform weighting) while keeping intermediate-layer alignment? This isolates the attention weighting contribution.

5. **Learned correspondences**: Do the attention maps (Figure 3) follow a structured pattern (e.g., diagonal, block structure)? What does this reveal about teacher-student layer relationships?

6. **Comparison with recent work**: Can you include comparisons with SA-MLP and KDGCL (cited but not compared)?

7. **Computational complexity**: What is the per-iteration cost of STRIDE vs. G-CRD? For n=1.9M (OGBN-Mag), computing D_str via G-CRD seems expensive—how is this handled?

8. **Graph-level tasks**: Does STRIDE work for graph classification or only node classification?

---

### Official Review · Reviewer_xnvr · 2025-10-31

**Soundness:** 3
**Presentation:** 3
**Contribution:** 3
**Rating:** 6
**Confidence:** 4

**Summary:**

This paper introduces STRIDE, a new knowledge distillation framework for compressing GNNs. The key idea is to use a learnable attention mechanism to dynamically match intermediate layers between the teacher and student, instead of relying only on the final layer outputs as in most prior work. STRIDE distills both structural and node embedding information and includes subspace projections so that teacher and student layers with different dimensions can be aligned. Experiments on large benchmarks such as OGBN-Mag and OGBN-Arxiv, as well as smaller citation datasets, show consistent improvements over state-of-the-art GNN KD baselines, especially under high compression ratios. The paper also provides ablations and a theoretical discussion supporting the learning dynamics.

**Strengths:**

1. The paper introduces a trainable attention mechanism to dynamically distill knowledge from all intermediate layers. This resolves the key challenge of aligning teacher and student models with different architectures (e.g., varying depths) without requiring fixed layer correspondence.
2. The method is the first attention-based GNN KD framework to simultaneously align both graph structure and node embeddings. Ablation studies confirm that aligning both is superior to aligning only one.
3. The approach demonstrates superior accuracy on large-scale, out-of-distribution benchmarks like OGBN-Mag and achieves extreme compression ratios on smaller datasets like Pubmed while maintaining high accuracy.

**Weaknesses:**

1. Distillation cost and memory overhead from layer-wise projections and attention are not analyzed; may limit applicability to very large GNNs.
2. The method requires separate, trainable linear projection layers for every teacher and student layer (for attention, structure, and embedding alignment). This adds significant parameter overhead and computational complexity during the training phase.
3. Empirical scope is limited to node classification; no results for link prediction, inductive graphs, dynamic graphs, or molecular domains.

**Questions:**

1.	What is the training overhead (in terms of wall-clock time, GPU memory, and convergence speed) of STRIDE compared to baselines like G-CRD and LSP, given the added complexity of computing the $T_l \times S_l$ matrices and the multiple trainable projection layers?
2.	The method relies on "average node features" to compute inter-layer attention. Is this simple aggregation mechanism sufficient to capture complex layer-wise semantics for accurate alignment, especially on large heterogeneous graphs like OGBN-Mag? Were more expressive graph pooling methods explored?
3.	The paper claims G-CRD provides the best results for the structural dissimilarity component over LSP or GSP, but the supporting ablation study appears to be missing. Could the authors provide the experimental comparison of STRIDE variants using LSP and GSP for this component?

---

### Official Review · Reviewer_GwmG · 2025-11-04

**Soundness:** 3
**Presentation:** 3
**Contribution:** 3
**Rating:** 4
**Confidence:** 2

**Summary:**

The paper is motivated by the observation that most GNN knowledge distillation (KD) methods only align the final layer. The authors argue that this ignores the rich inductive biases about graph structure and node embeddings found in GNN intermediate layers, which can lead to insufficient learning for the student model. This paper aims to align the intermediate layers between a large teacher GNN and a compact student GNN.
To this end, the paper proposes STRIDE, a framework that uses a trainable attention mechanism to dynamically identify and align salient teacher-student layer pairs, matching both structural and embedding information. Ablation studies support the method's design, demonstrating that simultaneously aligning both structure and embeddings (full STRIDE) is significantly more effective than aligning only structure (S-STRIDE) or only embeddings (E-STRIDE).

**Strengths:**

The paper addresses a limitation in GNN compression by moving beyond final-layer KD. The core idea of using a new attention mechanism to dynamically match layers. Ablation studies provide evidence for the method's key design choices.

**Weaknesses:**

While the core objective is to compress GNNs to reduce inference costs, the STRIDE method itself introduces overhead during the training phase.
The method requires equipping each teacher and student layer with multiple sets of trainable linear projection layers (for attention, embedding dissimilarity, and structural dissimilarity), as well as an additional subspace projection matrix $P$.
In essence, STRIDE represents a strategy of "using high training costs to achieve excellent inference gains," but the extent of this extra training cost is not fully quantified.

**Questions:**

1.  Theorem 1 seems too simple to be a "Theorem." Its proof is primarily a straightforward application of the chain rule. It might be better to present this as a "Proposition."

2.  STRIDE is indeed a form of intermediate-layer distillation, specifically an "all-to-all" dynamic matching. It computes the knowledge gap between all teacher layers and all student layers, then uses an attention mechanism $\alpha_{ij}$ to dynamically decide which pairs are most important to align. The authors have effectively shown that STRIDE is superior to baseline methods that only align the last layer. However, to more clearly demonstrate the value of the *dynamic attention* component itself, the authors may compare it against simpler, less computationally expensive, *non-dynamic* intermediate-layer alignment strategies? For example:

    * Strategy A (Many-to-One): Distill knowledge from all teacher intermediate layers into a single student layer (e.g., the final layer).
    * Strategy B (One-to-Many): Distill knowledge from a single teacher layer (e.g., the final layer) to all student intermediate layers.

---

### Note · Authors · 2025-12-06

I have read and agree with the venue's withdrawal policy on behalf of myself and my co-authors.